# Facile Synthesis of 2*H*-Benzo[*h*]Chromenes via an Arylamine-Catalyzed Mannich Cyclization Cascade Reaction

**DOI:** 10.3390/molecules26123617

**Published:** 2021-06-12

**Authors:** Yueteng Zhang, Peng Ji, Xiang Meng, Feng Gao, Fanxun Zeng, Wei Wang

**Affiliations:** 1Departments of Pharmacology and Toxicology and Chemistry and Biochemistry, and BIO5 Institute, University of Arizona, Tucson, AZ 85721, USA; pji@pharmacy.arizona.edu (P.J.); xmeng@pharmacy.arizona.edu (X.M.); fgao@pharmacy.arizona.edu (F.G.); fzeng@pharmacy.arizona.edu (F.Z.); 2The School of Basic Medical Sciences, The Academy of Medical Science, Zhengzhou University, 100 Kexue Avenue, Zhengzhou 450001, China

**Keywords:** aminocatalysis, cascade reaction, chromenes, organocatalysis, quinone methide

## Abstract

A simple arylamine-catalyzed Mannich-cyclization cascade reaction was developed for facile synthesis of substituted 2*H*-benzo[*h*]chromenes. The notable feature of the process included the efficient generation of *ortho*-quinone methides (*o*-QMs) catalyzed by a simple aniline. The mild reaction conditions allowed for a broad spectrum of 1- and 2-naphthols and *trans*-cinnamaldehydes to engage in the cascade sequence with high efficiency.

## 1. Introduction

2*H*-Benzo[*h*]chromenes have attracted considerable attention in recent years due to their applications in medicinal chemistry and drug discovery, optical devices, and molecular imaging [1,2,3,4,5,6]. In particular, they have been extensively studied in photochromism [7,8,9,10]. The structures can undergo ring-opening with a color change via the photochemical cleavage of the sp^3^ C–O bond. These unique optical properties arise from their reversible changes through irradiation with ultraviolet or visible light with excellent photochromic responses, good colorability, and rapid bleaching. The broad distribution of the naphthopyans in numerous naturally and biologically active compounds [1,2,3,4,5,6], and their predominant applications in optical devices [7,8,9,10], have made the molecular architecture an attractive target for synthesis [2,3].

Along with the classical approaches, significant efforts have been devoted to developing more efficient catalytic methods, largely with transition metals [2,3]. In contrast, there are only a handful of reports about the synthesis of the 2*H*-benzo[*h*]chromenes. The methods are dominated by the acid-catalyzed condensation of naphthols with propargylic alcohol by *p*-toluenesulfonic acid [11,12], acidic alumina [13], indium trichloride [14], and montmorillonite K-10 [15]. Nucleophilic addition dehydration reactions of chromanones with Grignard [16] or organolithium [17] reagents provide an alternative. Cycloaddition reactions of naphthols with simple α, β-unsaturated aldehydes are also attractive strategies. However, the reactions are performed under harsh (refluxing) reaction conditions [18,19,20,21,22]. New methods with mild reaction conditions will streamline the synthesis of 2*H*-benzo[*h*]chromenes, which can tolerate broad functional groups. Herein, we wish to document a simple arylamine catalyzed formation of the molecular architectures under mild reaction conditions with a broad substrate scope (Scheme 1). A new reactivity involving the aniline-catalyzed in situ formation of *ortho*-quinone methide (*o*-QM) in a catalytic fashion is proposed. The reactivity is harnessed for developing a synthetically powerful Mannich–Diels–Alder cascade manifold for the construction of 2*H*-benzo[*h*]chromenes.

## 2. Results and Discussion

### 2.1. Design Plan

*o*-QM, a highly reactive intermediate originally harnessed in nature for various biological purposes [23], has been slowly recognized by synthetic organic chemists in organic synthesis [24,25,26,27,28,29]. Given the high reactivity of *o*-QMs, their in situ formation from the corresponding stable aromatic precursors by temporarily masking the “O” is generally carried out [24,25]. Therefore, an additional activation step is often required. In addition, the synthesis of the precursors is not trivial. We propose a new, more efficient catalytic approach for the in situ generation of *o*-QMs from readily accessible simple α, β-unsaturated aldehydes (enals) (Scheme 1). We envision that the direct functionalization of the aldehyde group of enals **1** with an amine catalyst gives imine **4**. The reaction of **4** with 1-naphthol **2** via a Mannich-type leads to product **5**, which serves as a precursor for a subsequent *o*-QM (**6**) formation by extrusion of the amine catalyst. Finally, an intramolecular cyclization reaction delivers benzopyrans, driven by re-aromatization.

Although the proposed work looks straightforward on paper, there are several key barriers to overcome for achieving the desired cascade transformations. First, the 1,2-addition reaction (Mannich) of **1** with 1-naphthols **2** may compete with the possible 1,4-conjuage addition process, which has been observed in chiral pyrrolidine-catalyzed Michael cyclization [30]. This suggests that the amine catalysts play key roles in the fate of the products formed. To minimize the 1,4-conjuage addition process, weakly nucleophilic, less-hindered aromatic amines may be the choice of promoters for achieving the desired 1,2-addition. It is believed that the aniline-derived imine **4** favors the 1,2- over 1,4-addition reaction, because its less-electrophilic and more-hindered features reduce the possibility of 1,4-addition, while the less-steric hindrance enhances the 1,2-addition accessibility. Second, achieving an unprecedented catalytic process for the formation of *o*-QMs is highly appealing but formidably challenging. Precedent studies relied on the preformation of the precursors using a stoichiometric amount of amine, and the generation of *o*-QMs requires light or acid activation [31,32]. We contemplated that the better leaving tendency of aniline may enable the process in a catalytic manner. The nucleophilicity and leaving tendency of amine need to be balanced to catalyze this cascade process effectively. Due to the better leaving ability of arylamines compared with aliphatic amines, we successfully developed aniline-promoted cyclization replacement cascade reactions of 2-hydroxycinnamaldehydes with various carbonic nucleophiles for the synthesis of dihydrobenzopyrans [33].

### 2.2. Exploration

To test the possibility of in situ-generated *o*-QMs promoted by aniline, followed by an intramolecular Diels–Alder reaction, a reaction between *trans*-cinnamaldehyde (**1a**) and 1-naphthol (**2a**) was conducted in the presence of a catalytic amount of an arylamine (20 mol%; Table 1). Gladly, the reaction with 20 mol% of **I** proceeded to give the desired product 2-phenyl-2*H*-benzo[*h*]chromene (**3a**), albeit a low yield of 10% after 48 h. Encouraged by the results, other anilines containing electron-donating or -withdrawing substituents were probed. 2-Methyoxyaniline (**III**) was much better than 4-methyoxyaniline (**II**) (entry 2 vs. 3) for this cascade process, whereas an electron-deficient group like 4-nitroaniline (**IV**) was less effective (entry 4). Naphthalen-1-amine (**V**) and a secondary aniline 1,2,3,4-tetrahydroquinoline (**VI**) were employed, showing an extremely low catalytic potency (entries 5 and 6). Then, we thought about whether *ortho*-hydrogen-bonding donor groups in the catalyst could accelerate this reaction. Thus, the 2-aminophenol (**VII**), 2-aminobenzoic acid (**VIII**) and *o*-phenylenediamine (**IX**) aniline analogs were studied (entries 7–9). **IX** gave the highest yield of 56% (entry 9). Therefore, we chose **IX** as the choice for further optimization.

The solvent effect on the reaction was probed next (Table 1, entries 18–24). Et_2_O and THF produced low-yielding products (entries 18 and 19). The same happened with polar solvents MeOH and MeCN (entries 20 and 21). This reaction was favored by nonpolar aprotic reaction media (entries 9 and 22–24). Further studies were carried out to optimize the Mannich cyclization cascade reaction in CH_2_Cl_2_ with catalyst **IX** (Table 1). Even though the reaction time was prolonged and the temperature was raised to 40 °C, no further improvement was attended (entries 10 and 11). Increasing the amount of **2a** to 1.5 equiv. gave better results (entry 12). When 1-naphthol (**2a**) was used as a limiting reactant, significant increase in yield (65%) was observed (entry 13). Furthermore, higher temperature, it not only reduced the reaction time but also enhanced the efficiency (entry 14). Finally, when the reaction was performed in dry DCM under Ar atmosphere, the yield was further improved, suggesting a negative effect of water in the reaction (87%, entry 15). A decreased catalyst loading lowered its efficiency (entry 16). Interestingly, *p*-phenylenediamine (**X**) gave a very low yield, which indicates that *ortho*-diamine plays a critical role in catalysis (entry 17). Therefore, the optimal reaction conditions for the formation of **3a** involved the usage of 0.3 mmol of **1a**, 0.2 mmol of **2a** and 20 mol% of **IX** in 1 mL of dry DCM under Ar.

### 2.3. Reaction Scope 

An investigation was carried out to probe the scope of the reactions of the substituted *trans*-cinnamaldehyde **1** with 1-naphthol (**2**) (Scheme 2). It seems the process served as a general approach for the production of structurally diverse 2*H*-benzo[*h*]chromenes. The reaction was less sensitive to electronic and steric effects. The enals with electron-withdrawing groups such as nitro, chloro, floro and trifloromethyl all gave rise to good yields (**3b**–**d**, **3g**). A similar trend was observed for *trans*-cinnamaldehydes bearing electron-donating groups, which furnished 2*H*-benzo[*h*]chromenes **3h**–**l** in high yields (**3h**–**l**). The steric demand produced a minimal impact as well (**3b**, **3f**, **3g** and **3i**–**l**). Notably, the mild protocol tolerated various functional groups, such as MeO, hydroxyl, acetoxy, NMe_2_, etc. Next, we probed the structural variations of 1-napthnols (Scheme 1). The protocol could tolerate electron-withdrawing groups such as Cl, Br, NO_2_, OMe, NHBoc and NHAc to give the corresponding products **3m**–**r** in good yields. Finally, the heteroaromatic enals, including furan, pyridine and quinolone, were examined. They participated in the process to smoothly deliver the desired products **3s**–**u** with high efficiency.

Since 2,2-disubstiuted-2*H*-chromenes are a common motif in many natural products and bioactive compounds and photochromic molecular devices, we made efforts toward their synthesis using this approach (Scheme 3). Gladly, 2,2-dimethyl-2*H*-benzo[*h*]chromene **3v** was synthesized with a good yield of 61%. Similarly, when 3,3-diphenylacrylaldehyde was served as a substrate, the corresponding product **3w** was obtained with a 78% yield. To further explore the versatility of this new synthetic strategy, reactions by employing 2-naphthols with enal **1** were performed. Both processes ran smoothly with good yields (**3x** and **3y**). When 1,5-dihydroxynaphthalene was served as a substrate, a bis-cyclized product 3,9-diphenyl-3,9-dihydrochromeno[8,7-*h*]chromene (**3z**) was formed efficiently in a one-pot operation. Polyhydroxy resorcinol also led to the bis product 2,8-diphenyl-2*H*,8*H*-pyrano[2,3-*f*]chromene (**3ab**) in a good yield of 60% in a sealed tube at 60 °C. In contrast, when 2,7-dihydroxynaphthalene (**2v**), which also has two reaction sites, was employed, only mono-product **3aa** was formed with an excellent yield of 95%. Finally, the method was demonstrated as a powerful approach for the efficient synthesis of natural product lapachenole (**3ac**) [34].

## 3. Materials and Methods

All commercially available reagents were purchased from Sigma-Aldrich, TCI, Alfa, Ambeed in the US and were used without further purification. All reactions were carried out in dried flasks. The progress of the reactions was monitored by analytical thin-layer chromatography (TLC) on Whatman silica gel plates with a fluorescence F254 indicator. Merck 60 silica gel was used for chromatography. ^1^H (300 MHz) and ^13^C (75 MHz) NMR spectra were recorded on a Bruker Avance 300 (Appendix A). When deuterated chloroform (CDCl_3_) was used, residue chloroform was used as an internal reference. When deuterated dimethyl sulfoxide (DMSO) was used, residue dimethyl sulfoxide was used as an internal reference. Data for ^1^H NMR were reported as follows: chemical shift (ppm) and multiplicity (s = singlet, d = doublet, t = triplet, q = quartet and m = multiplet). Chemical shift for ^13^C NMR was reported as ppm.

## 4. General Procedure for the Synthesis of 2*H*-Benzo[*h*]Chromene 3

Dry CH_2_Cl_2_ (1.0 mL) was added to a mixture of *o*-phenylenediamine (**IX**, 0.04 mmol), enal **1** (0.30 mmol) and arylol **2** (0.20 mmol) under Ar. The reaction was stirred at 40 °C until the completion of **2** monitored by TLC. Then, the mixture was applied to column chromatography directly and eluted with ethyl ether and hexane (9/1) to give product **3**.

2-Phenyl-2*H*-benzo[*h*]chromenes (**3a**). The title compound was prepared according to the general procedure: 17 h, red oil; 87% yield. ^1^H NMR (300 MHz, CDCl_3_) δ (ppm) 8.27–8.24 (m, 1H), 7.81–7.76 (m, 1H), 7.60–7.56 (m, 2H), 7.48–7.39 (m, 6H), 7.22 (d, *J* = 8.4 Hz, 1H), 6.70 (dd, *J* = 9.8, 1.7 Hz, 1H), 6.18 (dd, *J* = 3.6, 1.7 Hz, 1H), 5.92 (dd, *J* = 9.8, 3.6 Hz, 1H). ^13^C NMR (75 MHz, CDCl_3_) δ (ppm) 148.65, 141.21, 134.74, 128.73, 128.35, 127.71, 126.86, 126.41, 125.56, 124.71, 124.64, 123.39, 122.11, 120.51, 115.80, 77.34. HRMS (ESI) calcd for [C_19_H_14_O + H]^+^: 259.1123, found: 259.1128.

2-(2-Nitrophenyl)-2*H*-benzo[*h*]chromene (**3b**). The title compound was prepared according to the general procedure: 19 h, orange oil; 72% yield; ^1^H NMR (300 MHz, CDCl_3_) δ (ppm) 8.15–8.11 (m, 1H), 8.02 (dd, *J* = 8.1, 1.2 Hz, 1H), 7.87 (dd, *J* = 7.9, 1.2 Hz, 1H), 7.79–7.74 (m, 1H), 7.56 (td, *J* = 7.5, 1.2 Hz, 1H), 7.48–7.39 (m, 4H), 7.17 (d, *J* = 8.4 Hz, 1H), 6.84 (dd, *J* = 3.6, 1.8 Hz, 1H), 6.66 (dd, *J* = 9.9, 1.8 Hz, 1H), 5.92 (dd, *J* = 9.9, 3.6 Hz, 1H). ^13^C NMR (75 MHz, CDCl_3_) δ 148.40, 147.09, 136.89, 134.84, 133.82, 128.91, 128.86, 127.82, 126.66, 125.90, 125.13, 124.87, 124.79, 124.28, 121.86, 121.78, 121.03, 115.27, 73.24. HRMS (ESI) calcd for [C_19_H_13_NO_3_ + H]^+^: 304.0974, found: 304.0971.

2-(4-Nitrophenyl)-2*H*-benzo[*h*]chromene (**3c**)**.** The title compound was prepared according to the general procedure except that 50 mg 4 Å MS powder was added: 60 h, orange oil; 83% yield; ^1^H NMR (300 MHz, CDCl_3_) δ (ppm) 8.21–8.16 (m, 3H), 7.78–7.72 (m, 1H), 7.66 (d, *J* = 8.7 Hz, 2H), 7.49–7.39 (m, 3H), 7.17 (d, *J* = 8.4 Hz, 1H), 6.71 (dd, *J* = 9.9, 1.8 Hz, 1H), 6.19 (dd, *J* = 3.9, 1.8 Hz 1H), 5.88 (dd, *J* = 9.9, 3.9 Hz, 1H). ^13^C NMR (75 MHz, CDCl_3_) δ 148.16, 148.08, 147.81, 134.86, 127.88, 127.35, 126.76, 125.97, 125.61, 124.65, 124.44, 124.00, 121.82, 121.75, 121.18, 115.67, 75.93. HRMS (ESI) calcd for [C_19_H_13_NO_3_ + H]^+^: 304.0974, found: 304.0969.

2-(4-Fluorophenyl)-2*H*-benzo[*h*]chromene (**3d**). The title compound was prepared according to the general procedure: 46 h; red oil; 83% yield; ^1^H NMR (300 MHz, CDCl_3_) δ (ppm) 8.18–8.15 (m, 1H), 7.76–7.73 (m, 1H), 7.53–7.48 (m, 2H), 7.46–7.37 (m, 3H), 7.18 (d, *J* = 8.1 Hz, 1H), 7.09–7.02 (m, 2H), 6.69 (dd, *J* = 9.9, 1.8 Hz, 1H), 6.11 (d, *J* = 2.4 Hz, 1H), 5.86 (dd, *J* = 9.6, 3.6 Hz, 1H). ^13^C NMR (75 MHz, CDCl_3_) δ (ppm) 162.82 (d, *J* = 245 Hz), 148.38, 136.87, 134.76, 128.91, 128.80, 127.76, 126.50, 125.65, 124.96, 124.68, 123.02, 122.00, 120.66, 115.76, 115.48, 76.58. HRMS (ESI) calcd for [C_19_H_13_FO + H]^+^: 277.1029, found: 277.1032.

2-(4-Chlorophenyl)-2*H*-benzo[*h*]chromene (**3e**). The title compound was prepared according to the general procedure: 17 h; red oil; 81% yield; ^1^H NMR (300 MHz, CDCl_3_) δ (ppm) 8.21–8.16 (m, 1H), 7.78–7.74 (m, 1H), 7.49–7.39 (m, 5H), 7.36–7.32 (m, 2H), 7.19 (d, *J* = 8.4 Hz, 1H), 6.69 (dd, *J* = 9.9, 1.8 Hz, 1H), 6.10 (dd, *J* = 3.9, 1.8 Hz, 1H), 5.85 (dd, *J* = 9.9, 3.9 Hz, 1H). ^13^C NMR (75 MHz, CDCl_3_) δ 148.32, 139.55, 134.75, 134.21, 128.90, 128.33, 127.77, 126.53, 125.70, 125.03, 124.67, 124.60, 122.75, 121.95, 120.73, 115.74, 76.44. HRMS (ESI) calcd for [C_19_H_13_ClO + H]^+^: 293.0733, found: 293.0736.

2-(2-Bromophenyl)-2*H*-benzo[*h*]chromene (**3f**). The title compound was prepared according to the general procedure: 66 h; red oil; 85% yield; ^1^H NMR (300 MHz, CDCl_3_) δ (ppm) 8.24–8.19 (m, 1H), 7.79–7.74 (m, 1H), 7.66–7.62 (m, 2H), 7.48–7.39 (m, 3H), 7.29 (td, *J* = 7.5, 0.9 Hz, 1H), 7.19 (d, *J* = 8.1 Hz, 2H), 6.68 (dd, *J* = 9.9, 1.8 Hz, 1H), 6.53 (dd, *J* = 3.6, 1.8 Hz, 1H), 5.91 (dd, *J* = 9.9, 3.6 Hz, 1H). ^13^C NMR (75 MHz, CDCl_3_) δ (ppm) 148.68, 140.00, 134.78, 133.18, 129.75, 128.92, 127.93, 127.72, 126.53, 125.71, 124.97, 124.69, 124.54, 122.19, 122.14, 121.85, 120.70, 115.50, 76.46. HRMS (ESI) calcd for [C_19_H_13_BrO + H]^+^: 337.0228, found: 337.0224.

2-(2-(Trifluoromethyl)phenyl)-2*H*-benzo[*h*]chromene (**3g**). The title compound was prepared according to the general procedure: 42 h, red oil; 71% yield; ^1^H NMR (300 MHz, CDCl_3_) δ (ppm) 8.15–8.12 (m, 1H), 7.95 (d, *J* = 7.8 Hz, 1H), 7.78–7.75 (m, 2H), 7.55 (t, *J* = 7.5 Hz, 1H), 7.47–7.39 (m, 4H), 7.21 (d, *J* = 8.4 Hz, 1H), 6.66 (dd, *J* = 9.6, 1.8 Hz, 1H), 6.61 (s, 1H), 5.75 (dd, *J* = 9.9, 3.3 Hz, 1H). ^13^C NMR (75 MHz, CDCl_3_) δ (ppm) 148.59, 139.92, 134.89, 132.57, 129.64, 128.43, 127.71, 126.74 (q, *J* = 30 Hz), 126.59, 125.97 (q, *J* = 2.1 Hz), 125.68, 124.69, 124.65, 123.03, 122.14, 120.74, 115.10, 73.71. HRMS (ESI) calcd for [C_20_H_13_F_3_O + H]^+^: 327.0997, found: 327.0999.

2-(4-Methoxyphenyl)-2*H*-benzo[*h*]chromene (**3h**). The title compound was prepared according to the general procedure: 17 h; red oil; 84% yield; ^1^H NMR (300 MHz, CDCl_3_) δ (ppm) 8.20–8.17 (m, 1H), 7.76–7.72 (m, 1H), 7.49–7.37 (m, 5H), 7.19 (d, *J* = 8.4 Hz,1H), 6.93–6.88 (m, 2H), 6.69 (dd, *J* = 9.8, 1.8 Hz, 1H), 6.10 (dd, *J* = 3.9, 1.8 Hz, 1H), 5.88 (dd, *J* = 9.8, 3.6 Hz, 1H), 3.79 (s, 3H). ^13^C NMR (75 MHz, CDCl_3_) δ (ppm) 159.82, 148.58, 134.73, 133.23, 128.56, 127.68, 126.35, 125.48, 124.76, 124.67, 123.44, 122.13, 120.39, 115.85, 114.11, 77.02, 55.36. HRMS (ESI) calcd for [C_20_H_16_O_2_ + H]^+^: 289.1229, found: 289.1235.

2-(2-Methoxyphenyl)-2*H*-benzo[*h*]chromene (**3i**). The title compound was prepared according to the general procedure: 24 h; red oil; 80% yield; ^1^H NMR (300 MHz, CDCl_3_) δ (ppm) 8.22–8.18 (m, 1H), 7.76–7.53 (m, 1H), 7.51 (dd, *J* = 7.5, 1.5 Hz, 1H), 7.45–7.39 (m, 2H), 7.37 (d, *J* = 8.1 Hz, 1H), 7.31–7.25 (m, 1H), 7.16 (d, *J* = 8.4 Hz, 1H), 6.95–6.89 (m, 2H), 6.58 (dd, *J* = 9.6, 1.8 Hz, 1H), 6.53 (dd, *J* = 3.6, 1.8 Hz, 1H), 5.90 (dd, *J* = 9.9, 3.6 Hz, 1H). ^13^C NMR (75 MHz, CDCl_3_) δ (ppm) 155.82, 149.02, 134.71, 129.81, 129.14, 127.73, 127.52, 126.28, 125.50, 124.80, 124.62, 123.78, 123.73, 122.16, 120.94, 120.31, 115.66, 110.70, 72.54, 55.65. HRMS (ESI) calcd for [C_20_H_16_O_2_ + H]^+^: 289.1229, found: 289.1230.

4-(2*H*-Benzo[*h*]chromen-2-yl)-*N,N*-dimethylaniline (**3j**). The title compound was prepared according to the general procedure, except that the mixture was stirred at room temperature: 36 h; purple oil; 54% yield; ^1^H NMR (300 MHz, CDCl_3_) δ (ppm) 8.18–8.14 (m, 1H), 7.3–7.70 (m, 1H), 7.43–7.33 (m, 5H), 7.18 (d, *J* = 8.4 Hz, 1H), 6.73–6.66 (m, 3H), 6.06 (dd, *J* = 3.9, 1.8 Hz, 1H), 5.87 (dd, *J* = 9.6, 3.6 Hz, 1H), 2.94 (s, 6H). ^13^C NMR (75 MHz, CDCl_3_) δ (ppm) 150.80, 148.73, 134.65, 128.56, 128.48, 127.61, 126.22, 125.33, 124.84, 124.72, 124.43, 123.73, 122.27, 120.13, 115.92, 112.43, 77.41, 40.59. HRMS (ESI) calcd for [C_21_H_19_NO + H]^+^: 302.1545, found: 302.1544.

4-(2*H*-Benzo[*h*]chromen-2-yl)-2-methoxyphenyl acetate (**3k**). The title compound was prepared according to the general procedure: 46 h; red oil; 99% yield; ^1^H NMR (300 MHz, CDCl_3_) δ (ppm) 8.20–8.17 (m, 1H), 7.75–7.72 (m, 1H), 7.44–7.36 (m, 3H), 7.19–7.16 (m, 2H), 7.10–7.00 (m, 2H), 6.66 (dd, *J* = 9.9, 1.8 Hz, 1H), 6.10 (dd, *J* = 3.6, 1.8 Hz, 1H), 5.87 (dd, *J* = 9.6, 3.6 Hz, 1H), 3.79 (s, 3H), 2.31 (s, 3H). ^13^C NMR (75 MHz, CDCl_3_) δ (ppm) 169.15, 151.28, 148.54, 140.16, 139.70, 134.76, 127.78, 126.50, 125.67, 124.91, 124.68, 124.62, 123.25, 122.93, 122.01, 120.71, 119.16, 115.85, 111.09, 76.97, 55.99, 20.81. HRMS (ESI) calcd for [C_22_H18O_4_ + H]^+^: 347.1283, found: 347.1283.

4-(2*H*-benzo[*h*]chromen-2-yl)-2-methoxyphenol (**3l**). The title compound was prepared according to the general procedure: 68 h; red oil; 77% yield; ^1^H NMR (300 MHz, CDCl_3_) δ (ppm) 8.19–8.16 (m, 1H), 7.76–7.72 (m, 1H), 7.45–7.36 (m, 3H), 7.19 (d, *J* = 8.1 Hz, 1H), 7.09–7.01 (m, 2H), 6.91 (d, *J* = 8.1 Hz, 1H), 6.68 (dd, *J* = 9.9, 1.8 Hz, 1H), 6.06 (dd, *J* = 3.3, 1.5 Hz, 1H), 5.87 (dd, *J* = 9.9, 3.6 Hz, 1H), 5.68 (s, 1H), 3.84 (s, 3H). ^13^C NMR (75 MHz, CDCl_3_) δ 148.56, 146.69, 145.90, 134.70, 133.05, 127.71, 126.38, 125.50, 124.79, 124.67, 123.53, 122.06, 120.46, 120.42, 115.88, 114.39, 109.82, 77.32, 55.99. HRMS (ESI) calcd for [C_20_H_16_O_3_ + H]^+^: 305.1178, found: 305.1177.

6-Chloro-2-phenyl-2*H*-benzo[*h*]chromene (**3m**). The title compound was prepared according to the general procedure: 46 h; red oil; 81% yield; ^1^H NMR (300 MHz, CDCl_3_) δ (ppm) ^1^H NMR (300 MHz, CDCl_3_) δ 8.22–8.13 (m, 2H), 7.57–7.45 (m, 5H), 7.41–7.33 (m, 3H), 7.29 (s, 1H), 6.60 (dd, *J* = 9.9, 1.8 Hz, 1H), 6.13 (dd, *J* = 3.6, 1.8 Hz, 1H), 5.91 (dd, *J* = 9.9, 3.6 Hz, 1H). ^13^C NMR (75 MHz, CDCl_3_) δ (ppm) 147.64, 140.68, 131.40, 128.81, 128.58, 127.47, 126.93, 126.29, 125.74, 124.48, 124.17, 123.75, 123.43, 122.47, 116.09, 77.46. HRMS (ESI) calcd for [C_19_H_13_ClO + H]^+^: 293.0733, found: 293.0735.

6-Bromo-2-phenyl-2*H*-benzo[*h*]chromene (**3n**). The title compound was prepared according to the general procedure, except that 50-mg 4Å MS powder was added: 72 h; red oil; 60% yield; ^1^H NMR (300 MHz, CDCl_3_) δ (ppm) 8.21–8.09 (m, 2H), 7.57–7.43 (m, 5H), 7.41–7.30 (m, 3H), 6.60 (dd, *J* = 9.9, 1.5 Hz, 1H), 6.14 (dd, *J* = 3.3, 1.5 Hz, 1H), 5.90 (dd, *J* = 9.6, 3.6 Hz, 1H). ^13^C NMR (75 MHz, CDCl_3_) δ (ppm) 148.34, 140.66, 132.57, 128.83, 128.60, 128.09, 127.76, 127.13, 126.94, 126.31, 125.93, 124.15, 123.62, 122.49, 116.73, 113.45, 77.50. HRMS (ESI) calcd for [C_19_H_13_BrO + H]^+^: 337.0228, found: 337.0232.

6-Nitro-2-phenyl-2*H*-benzo[*h*]chromene (**3o**). The title compound was prepared according to the general procedure, except that 50-mg 4Å MS powder was added: 72 h; red oil; 62% yield; ^1^H NMR (300 MHz, CDCl_3_) δ (ppm) 8.74 (d, *J* = 8.7 Hz, 1H), 8.24 (d, *J* = 8.7 Hz, 1H), 8.18 (s, 1H), 7.68–7.52(m, 1H), 7.52–7.47(m, 3H), 7.43–7.33(m, 3H), 6.65 (dd, *J* = 9.9, 1.5 Hz, 1H), 6.26 (dd, *J* = 3.3, 1.5 Hz, 1H), 5.96 (dd, *J* = 9.9, 3.6 Hz, 1H). ^13^C NMR (75 MHz, CDCl_3_) δ (ppm) 154.19, 139.86, 138.92, 130.26, 129.02, 127.18, 127.10, 126.76, 125.01, 124.67, 124.56, 123.74, 122.78, 122.73, 113.63, 78.85. HRMS (ESI) calcd for [C_19_H_13_NO_3_ + H]^+^: 304.0974, found: 304.0975.

6-Methoxy-2-phenyl-2*H*-benzo[*h*]chromene (**3p**). The title compound was prepared according to the general procedure: 26 h; red oil; 60% yield; ^1^H NMR (300 MHz, CDCl_3_) δ (ppm) 8.19–8.16 (m, 2H), 7.51–7.53 (m, 2H), 7.50– 7.32 (m, 5H), 6.64 (dd, *J* = 9.9, 1.8 Hz, 1H), 6.54 (s, 1H), 6.07 (dd, *J* = 3.6, 1.5 Hz, 1H), 5.93 (dd, *J* = 9.9, 3.9 Hz, 1H). ^13^C NMR (75 MHz, CDCl_3_) δ (ppm) 149.77, 142.45, 141.21, 128.68, 128.26, 126.84, 126.34, 126.27, 125.81, 125.48, 125.05, 123.92, 122.02, 121.84, 115.39, 102.64,76.86, 55.88. HRMS (ESI) calcd for [C_20_H_16_O_2_ + H]^+^: 289.1229, found: 289.1233.

*tert*-Butyl (2-phenyl-2*H*-benzo[*h*]chromen-6-yl)carbamate (**3q**). The title compound was prepared according to the general procedure: 41 h; red oil; 81% yield; ^1^H NMR (300 MHz, CDCl_3_) δ 7.99 (d, *J* = 8.4 Hz, 1H), 7.84 (d, *J* = 7.5 Hz, 1H), 7.52–7.50 (m, 2H), 7.41–7.32 (m, 5H), 7.18 (d, *J* = 8.4 Hz, 1H), 6.81 (s, 1H), 6.64 (dd, *J* = 9.6, 1.5 Hz, 1H), 6.13 (dd, *J* = 3.3, 1.2 Hz, 1H), 5.90 (dd, *J* = 9.9, 3.9 Hz, 1H), 1.57 (s, 11H). ^13^C NMR (75 MHz, CDCl_3_) δ (ppm) 153.60, 149.11, 141.01, 132.92, 128.75, 128.41, 127.69, 126.81, 125.51, 125.33, 124.72, 124.26, 123.78, 119.35, 118.52, 115.84, 113.01, 80.75, 77.39, 28.50. HRMS (ESI) calcd for [C_24_H_23_NO_3_ + H]^+^: 374.1756, found: 374.1753.

*N*-(2-Phenyl-2*H*-benzo[*h*]chromen-6-yl)acetamide (**3r**). The title compound was prepared according to the general procedure: 72 h; red oil; 95% yield; ^1^H NMR (300 MHz, DMSO) δ (ppm) 9.84 (s, 1H), 7.90 (d, *J* = 8.1 Hz, 1H), 7.66–7.59 (m, 2H), 7.50–7.31 (m, 7H), 6.79 (d, *J* = 9.6 Hz, 1H), 6.23 (d, *J* = 3.3 Hz, 1H), 6.11 (dd, *J* = 9.6, 3.9 Hz, 1H), 2.17 (s, 3H). ^13^C NMR (75 MHz, DMSO) δ (ppm) 168.84, 147.59, 140.68, 133.77, 128.18, 128.55, 126.37, 125.32, 124.53, 124.32, 124.23, 123.52, 121.99, 118.26, 115.75, 115.22, 75.97, 23.45. HRMS (ESI) calcd for [C_21_H_17_NO_2_ + H]^+^: 316.1338, found: 316.1340.

2-(Furan-2-yl)-2*H*-benzo[*h*]chromene (**3s**). The title compound was prepared according to the general procedure, except that the mixture was stirred at room temperature: 24 h; red oil; 51% yield; ^1^H NMR (300 MHz, CDCl_3_) δ (ppm) 8.21–8.18 (m, 1H), 7.74–7.70 (m, 1H), 7.45–7.36 (m, 4H), 7.19 (d, *J* = 8.4 Hz, 1H), 6.74 (dd, *J* = 9.6, 1.2 Hz, 1H), 6.41 (d, *J* = 3.0 Hz, 1H), 6.30 (dd, *J* = 2.7, 1.8 Hz, 1H), 6.14 (d, *J* = 3.3 Hz, 1H), 5.89 (dd, *J* = 9.6, 4.2 Hz, 1H). ^13^C NMR (75 MHz, CDCl_3_) δ (ppm) 152.88, 148.20, 143.40, 134.67, 127.65, 126.45, 126.07, 125.57, 124.82, 124.69, 122.16, 120.72, 119.85, 115.88, 110.47, 109.55, 70.04. HRMS (ESI) calcd for [C_17_H_12_O_2_ + H]^+^: 249.0916, found: 249.0918.

3-(2*H*-Benzo[*h*]chromen-2-yl)-2-bromopyridine (**3t**). The title compound was prepared according to the general procedure: 24 h; red oil; 80% yield; ^1^H NMR (300 MHz, CDCl_3_) δ (ppm) 8.31 (dd, *J* = 4.8, 2.1 Hz, 1H), 8.17 (dd, *J* = 6.0, 3.6 Hz, 1H), 7.87 (dd, *J* = 7.8, 1.8 Hz, 1H), 7.76–7.33 (m, 1H), 7.46–7.39 (m, 3H), 7.23–7.16 (m, 2H), 6.70 (dd, *J* = 9.9, 1.8 Hz, 1H), 6.43 (dd, *J* = 3.6, 1.8 Hz, 1H), 5.90 (dd, *J* = 9.6, 3.6 Hz, 1H). ^13^C NMR (75 MHz, CDCl_3_) δ (ppm) 149.66, 148.23, 141.44, 137.22, 137.13, 134.83, 127.78, 126.73, 125.92, 125.61, 124.60, 124.37, 123.34, 121.96, 121.10, 120.95, 115.37, 75.38. HRMS (ESI) calcd for [C_18_H_12_BrNO + H]^+^: 338.0181, found: 338.0180.

4-(2*H*-Benzo[*h*]chromen-2-yl)isoquinoline (**3u**). The title compound was prepared according to the general procedure: 24 h; red oil; 91% yield; ^1^H NMR (300 MHz, CDCl_3_) δ 8.90 (d, *J* = 4.5 Hz, 1H), 8.27 (d, *J* = 8.4 Hz, 1H), 8.19 (d, *J* = 8.4 Hz, 1H), 8.12–8.09 (m, 1H), 7.77–7.73 (m, 2H), 7.68–7.60 (m, 2H), 7.45–7.36 (m, 3H), 7.21 (d, *J* = 8.4 Hz, 1H), 6.80–6.74 (m, 2H), 5.91 (dd, *J* = 9.6, 3.6 Hz, 1H). ^13^C NMR (75 MHz, CDCl_3_) δ (ppm) 150.55 (s), 148.76, 148.64, 145.01, 134.86, 130.57, 129.40, 127.80, 127.10, 126.70, 126.14, 125.88, 125.61, 124.70, 124.52, 123.62, 121.99, 121.51, 121.17, 119.23, 115.75, 73.67. HRMS (ESI) calcd for [C_22_H_15_NO + H]^+^: 310.1232, found: 310.1236.

2,2-Dimethyl-2*H*-benzo[*h*]chromene (**3v**). The title compound was prepared according to the general procedure, except that 50-mg 4 Å MS powder was added: 72 h; colorless oil; 61% yield; ^1^H NMR (300 MHz, CDCl_3_) δ (ppm) 8.24–8.21 (m, 1H), 7.76–7.73 (m, 1H), 7.48–7.43 (m, 2H), 7.35 (d, *J* = 8.4 Hz, 1H), 7.16 (d, *J* = 8.4 Hz, 1H), 6.46 (d, *J* = 9.6 Hz, 1H), 5.65 (d, *J* = 9.6 Hz, 1H), 1.54 (s, 6H). ^13^C NMR (75 MHz, CDCl_3_) δ (ppm) 148.20, 134.40, 129.26, 127.52, 126.04, 125.17, 125.03, 124.49, 122.77, 121.93, 119.78, 115.33, 76.75, 27.93. HRMS (ESI) calcd for [C_15_H_14_O + H]^+^: 211.1123, found: 211.1121.

2,2-Diphenyl-2*H*-benzo[*h*]chromene (**3w**). The title compound was prepared according to the general procedure, except that 50-mg 4 Å MS powder was added: 34 h; red oil; 78% yield; ^1^H NMR (300 MHz, CDCl_3_) δ (ppm) 8.44 (d, *J* = 7.2 Hz, 1H), 7.77 (d, *J* = 8.1 Hz, 1H), 7.61–7.46 (m, 6H), 7.40–7.26 (m, 7H), 7.21 (d, *J* = 8.1 Hz, 1H), 6.78 (d, *J* = 9.6 Hz, 1H), 6.25 (d, *J* = 9.6 Hz, 1H). ^13^C NMR (75 MHz, CDCl_3_) δ (ppm) 147.70, 145.16, 134.63, 128.12, 127.63, 127.45, 127.25, 126.82, 126.27, 125.54, 124.67, 124.50, 123.83, 121.98, 120.46, 115.42, 83.16. HRMS (ESI) calcd for [C_25_H_18_O + H]^+^: 335.1436, found: 335.1436.

3-Phenyl-3*H*-benzo[*f*]chromene (**3x**). The title compound was prepared according to the general procedure, except that 50-mg 4 Å MS powder was added: 6 days; white solid; 68% yield; ^1^H NMR (300 MHz, CDCl_3_) δ (ppm) 8.00 (d, *J* = 9.0 Hz, 1H), 7.76 (d, *J* = 9.0 Hz, 1H), 7.67 (d, *J* = 9.0 Hz, 1H), 7.54–7.48 (m, 3H), 7.42–7.26 (m, 4H), 7.29–7.26 (m, 1H), 7.10 (d, *J* = 9.0 Hz, 1H), 6.01–5.94 (m, 2H). ^13^C NMR (75 MHz, CDCl_3_) δ (ppm) 151.47, 140.76, 130.03, 129.87, 129.53, 128.79, 128.72, 128.56, 127.25, 126.83, 123.75, 123.61, 121.41, 120.42, 118.15, 114.29, 77.07. HRMS (ESI) calcd for [C_19_H_14_O + H]^+^: 259.1123, found: 259.1127.

8-Bromo-3-phenyl-3*H*-benzo[*f*]chromene (**3y**). The title compound was prepared according to the general procedure, except that 50-mg 4 Å MS powder was added: 5 days; white solid; 60% yield; ^1^H NMR (300 MHz, CDCl_3_) δ (ppm) 7.89 (d, *J* = 1.5 Hz, 1H), 7.83 (d, *J* = 9.0 Hz, 1H), 7.56–7.50 (m, 4H), 7.42–7.34 (m, 3H), 7.18 (d, *J* = 9.6 Hz, 1H), 7.09 (d, *J* = 8.7 Hz, 1H), 6.01–5.94 (m, 2H). ^13^C NMR (75 MHz, CDCl_3_) δ (ppm) 151.65, 140.47, 130.63, 130.57, 129.97, 128.84, 128.68, 128.50, 127.23, 124.17, 123.25, 119.96, 119.25, 117.43, 114.43, 77.16. HRMS (ESI) calcd for [C_19_H_13_BrO + H]^+^: 337.0228, found: 337.0229.

3,9-Diphenyl-3,9-dihydrochromeno[8,7-h]chromene (**3z**). The title compound was prepared according to the general procedure, except that 50-mg 4 Å MS powder was added: 5 days; red oil; 51% yield; ^1^H NMR (300 MHz, CDCl_3_) δ (ppm) 7.73 (dd, *J* = 8.4, 1.2 Hz, 2H), 7.55–7.52 (m, 4H), 7.42–7.30 (m, 6H), 7.12 (d, *J* = 9.0 Hz, 2H), 6.64 (dd, *J* = 9.6, 1.5 Hz, 2H), 6.14–6.10 (m, 2H), 5.88 (ddd, *J* = 9.6, 3.6, 2.4 Hz, 2H). ^13^C NMR (75 MHz, CDCl_3_) δ (ppm) 148.50, 141.19, 128.73, 128.34, 126.85, 125.60, 124.62, 124.30, 123.63, 123.55, 116.38, 116.30, 114.68, 77.29. HRMS (ESI) calcd for [C_28_H_20_O_2_ + H]^+^: 389.1542, found: 389.1545.

3-Phenyl-3*H*-benzo[*f*]chromen-9-ol (**3aa**). The title compound was prepared according to the general procedure, except that 50-mg 4 Å MS powder was added: 60 h; orange solid; 95% yield; ^1^H NMR (300 MHz, CDCl_3_) δ (ppm) 7.62 (d, *J* = 8.7 Hz, 1H), 7.55 (d, *J* = 8.7 Hz, 1H), 7.50 (d, *J* = 6.9 Hz, 2H), 7.39–7.35 (m, 3H), 7.32–7.28 (m, 2H), 7.05 (d, *J* = 9.9 Hz, 1H), 6.97–6.90 (m, 2H), 5.94 (s, 1H), 5.87 (dd, *J* = 9.9, 3.0 Hz, 1H). ^13^C NMR (75 MHz, CDCl_3_) δ (ppm) 154.92, 152.28, 140.92, 130.79, 129.92, 128.92, 128.69, 127.39, 125.08, 123.17, 120.60, 115.82, 115.63, 113.33, 104.31, 77.16. HRMS (ESI) calcd for [C_19_H_14_O_2_ + H]^+^: 275.1072, found: 275.1074.

2,8-Diphenyl-2*H*,8*H*-pyrano[2,3-*f*]chromene (**3ab**). The title compound was prepared according to the general procedure, except that the mixture was heated at 60 °C in sealed tubed with 50-mg 4 Å MS powder: 5 days; orange solid; 60% yield; ^1^H NMR (300 MHz, CDCl_3_) δ (ppm) 7.53–7.35 (m, 10H), 6.92–6.88 (m, 1H), 6.82 (dd, *J* = 8.1, 2.1 Hz, 1H), 6.52–6.48 (m, 1H), 6.41 (ddd, *J* = 8.1, 2.4, 0.6 Hz, 1H), 5.97 (dd, *J* = 3.3, 1.8 Hz, 1H), 5.90–5.87 (m, 1H), 5.80–5.71 (m, 2H). ^13^C NMR (75 MHz, CDCl_3_) δ (ppm) 154.04, 148.86, 148.79, 140.94, 140.65, 140.58, 128.56, 128.29, 128.19, 127.05, 127.03, 126.79, 123.96, 123.87, 123.33, 123.20, 121.75, 121.60, 118.45, 118.29, 115.03, 114.92, 109.98, 109.88, 108.53, 77.07, 76.88. HRMS (ESI) calcd for [C_24_H_18_O_2_ + H]^+^: 339.1385, found: 339.1384.

6-Methoxy-2,2-dimethyl-2*H*-benzo[*h*]chromene (**3ac**). The title compound was prepared according to the general procedure, with 50-mg 4Å MS powder added: 60 h; colorless; 56% yield; ^1^H NMR (300 MHz, CDCl_3_) δ (ppm) 8.16 (d, *J* = 8.1 Hz, 2H), 7.51–7.41 (m, 2H), 6.52 (s, 1H), 6.41 (d, *J* = 9.6 Hz, 1H), 5.66 (d, *J* = 9.6 Hz, 1H), 3.96 (s, 3H), 1.51 (s, 6H). ^13^C NMR (75 MHz, CDCl_3_) δ (ppm) 149.42, 142.08, 130.04, 126.19, 126.04, 125.59, 123.19, 121.97, 121.83, 114.90, 102.69, 76.33, 55.92, 27.70. HRMS (ESI) calcd for [C_16_H_16_O_2_ + H]^+^: 241.1229, found: 241.1233.

## 5. Conclusions

In conclusion, we designed and realized a simple arylamine-catalyzed Mannich cyclization cascade via in situ-generated *o*-QMs to the formation of substituted 2*H*-benzo[*h*]chromenes. The notable features of the process included the generation of *o*-QMs catalyzed by a simple aniline for the first time. The mild reaction conditions allowed for a broad spectrum of 1- and 2-napthols and *trans*-cinnamaldehydes to engage in the cascade sequence with high efficiency. The development of enantioselective cascade reactions for the formation of enantioenriched 2*H*-benzo[*h*]chromenes and expansion of the application of a new catalytic strategy in synthesis are being pursued in our laboratories.

## Data Availability

Data of the compounds are available from the authors.

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
