# Peer review of "Facile Synthesis of 2H-Benzo[h]Chromenes via an Arylamine-Catalyzed Mannich Cyclization Cascade Reaction"

_molecules, 2021, doi:10.3390/molecules26123617_

Round 1
Reviewer 1 Report
This manuscript by Wang et al reports on a systematic study of an arylamine-catalyzed cascade reaction between a series of 1- and 2-naphthols and trans-cinnamaldehydes under mild conditions to provide a series of substituted 2H-benzo[h]chromenes. The mechanism is hypothesized, based on literature precedence, to occur via an initial Mannich reaction, followed by an elimination to a quinone methide and subsequent electrocyclization reaction. The manuscript provides good background documentation with many references, and the experimental work is well described and supported with sufficient analytical data. This work is a direct extension of Lee et al (a) Tetrahedron Lett. 2005, 46, 7539 (b) Helvetica Chimica Acta, vol. 90 (2007) 2401 (cited as reference 22), in which the same type of cascade reaction is catalyzed by ethylenediamine diacetic acid salt in refluxing chloroform. These are mild conditions, contrary to the claims of “harsh conditions” by the current authors, and makes the current work less novel than claimed. Nevertheless, the many examples provided offer a mild and efficient complementary method for the synthesis of substituted 2H-benzo[h]chromenes and congeners. The paper is recommended for publication, but only after substantial changes, many minor, as follows:
- Abstract, line 10: Remove “unprecedented”; “Mannich-Cyclization” should be “Mannich-cyclization” ; line 12: “o-QMs” should be “ortho-quinone methides (o-QMs)”.
- Introduction, line 41: “ ortho-Quinone methide” should be “ortho-quinone methide”.
- Introduction, lines 42 – 43: the sentence “The reactivity transforms into an unprecedented synthetically powerful Mannich-Diels-Alder cascade manifold for the construction of 2H-benzo[h]chromenes”. This is not an entirely accurate statement. The prior work of Lee et al (reference 22) could also be viewed as a Mannich-Diels-Alder cascade. The authors should remove the word “unprecedented”. There are other places in the manuscript where this word should be removed.
- Introduction, Scheme 1: Much of the electron rearrangement scheme for the Mannich reaction is incorrect. The flow of electrons for the Mannich reaction between structures 2 and 4 should show 2 with an arrow from an electron pair of the oxygen phenol forming a protonated ketone in structure 5, and rearrangement of each of the double bonds of the phenol ring with attack of one double bond onto the preformed imine. This will give structure 5 as shown, but as a ketone and two double bonds in the left ring and a sp3 carbon attached to the exocyclic fragment. Elimination of the aniline will then generate structure 6, which is correctly drawn.
- Results and Discussion, lines 51 & 52: The sentence “Given their high reactivities, generally the “O” is temporarily masked as stable aromatic precursors [24,25]” is vague and unclear. Please modify to make clear what the authors intent to communicate.
- Results and Discussion, lines 67 & 68: the sentence “It is believed that the aniline derived imine 4 favors the 1,2- over 1,4-addition reaction because its less electrophilic and less hindered features reduce the electrophilicity of the β-position while the less steric demand enhances the 1,2-addition accessibility” is a cumbersome, confusing sentence. Please modify to make clearer what the authors intend to communicate.
- Table 1. Entries 9, 11 are the same. Entry 11 probably should be at reflux. There should also be some discussion on entries 12 – 17.
- Exploration, line 82: Change “D-A” to “Diels-Alder”
- Exploration, line 108 – 109: the sentence “Increasing the amount of 1b to 1.5 equiv. gave better results (entry 12)” appears to be incorrect There is no 1b. Table 1 says 2a. The authors need to carefully check that all the conditions shown in Table 1 are correct and align with discussion comments.
- Exploration, line 109. “1-naphonol” should be “1-naphthol”
- Exploration, line 110 – 111: “Furthermore, higher temperature not only reduced reaction time but also enhanced the efficiency (entry 14)”. The Table 1 entry shows the reaction run at room temperature.
- Materials and Methods. Since none of the NMRs were taken in deuterated DMSO, remove that sentence from the general description.
- General comments: the entire manuscript needs a careful re-reading and editing as there are several sentences that are unclear and a number of misspellings. Some of these have been noted above, but not all have been flagged.
Author Response
Reviewer 1
- “Abstract, line 10: Remove “unprecedented”; “Mannich-Cyclization” should be “Mannichcyclization”; line 12: “o-QMs” should be “ortho-quinone methides (o-QMs)”.”
Response: As suggested, “unprecedented” has been removed. We have replaced “Mannich-Cyclization” with “Mannich-cyclization”. “o-QMs” has been replaced with “ortho-quinone methides (o-QMs)”
- “Introduction, line 41: “ ortho-Quinone methide” should be “ortho-quinone methide”.”
Response: As suggested, “ ortho-Quinone methide” has been changed to “ortho-quinone
methide”
- “Introduction, lines 42 – 43: the sentence “The reactivity transforms into an unprecedented synthetically powerful Mannich-Diels-Alder cascade manifold for the construction of 2Hbenzo[h]chromenes”. This is not an entirely accurate statement. The prior work of Lee et al (reference 22) could also be viewed as a Mannich-Diels-Alder cascade. The authors should remove the word “unprecedented”. There are other places in the manuscript where this word should be removed.”
Response: As suggested, the word “unprecedented” has been removed in the manuscript.
- “Introduction, Scheme 1: Much of the electron rearrangement scheme for the Mannich reaction is incorrect. The flow of electrons for the Mannich reaction between structures 2 and 4 should show 2 with an arrow from an electron pair of the oxygen phenol forming a protonated ketone in structure 5, and rearrangement of each of the double bonds of the phenol ring with attack of one double bond onto the preformed imine. This will give structure 5 as shown, but as a ketone and two double bonds in the left ring and a sp3 carbon attached to the exocyclic fragment.
Elimination of the aniline will then generate structure 6, which is correctly drawn.”
Response: As suggested, Scheme 1 has been corrected.
- “Results and Discussion, lines 51 & 52: The sentence “Given their high reactivities, generally the “O” is temporarily masked as stable aromatic precursors [24,25]” is vague and unclear.
Please modify to make clear what the authors intent to communicate.”
Response: As suggested, we have revised the sentence, as follows: “Given the high reactivity of o-QMs, their in situ formation from the corresponding stable aromatic precursors by temporarily masking the “O” is generally carried out [24,25].”
- “Results and Discussion, lines 67 & 68: the sentence “It is believed that the aniline derived imine 4 favors the 1,2- over 1,4-addition reaction because its less electrophilic and less hindered features reduce the electrophilicity of the β-position while the less steric demand enhances the 1,2-addition accessibility” is a cumbersome, confusing sentence. Please modify to make clearer what the authors intend to communicate.”
Response: We have modified the sentence as “It is believed that the aniline derived imine 4 favors the 1,2- over 1,4-addition reaction because its less electrophilic and less hindered
features reduce the possibility of 1,4-addition while the less steric hindrance enhances the 1,2-addition accessibility.”
- “Table 1. Entries 9, 11 are the same. Entry 11 probably should be at reflux. There should also be some discussion on entries 12 – 17.”
Response: This is typo in entry 11. It has been replaced by “40”. And the discussion has been added into the text, as follow: “Finally, when the reaction was performed in dry DCM under Ar atmosphere, the yield was further improved suggesting the negative effect of water in reaction (87%, entry 15). Decreased catalyst loading lowered its efficiency (entry 16). Interestingly, using p-phenylenediamine (X) gave very low yield, which indicates orthodiamine plays an important role in catalyis (entry 17).”
- “Exploration, line 82: Change “D-A” to “Diels-Alder””
Response: As suggested, “D-A” has been changed to “Diels-Alder”.
- “Exploration, line 108 – 109: the sentence “Increasing the amount of 1b to 1.5 equiv. gave better results (entry 12)” appears to be incorrect There is no 1b. Table 1 says 2a. The authors need to carefully check that all the conditions shown in Table 1 are correct and align with discussion comments.”
Response: As suggested, this sentence has been corrected as “Increasing the amount of 2a to 1.5 equiv. gave better results (entry 12). When 1-naphthol (2a) was used as a limiting reactant, significant increase in yield (65%) was observed (entry 13).”
- “Exploration, line 109. “1-naphonol” should be “1-naphthol””
Response: As suggested, “1-naphonol” has been replaced by “1-naphthol”.
- “Exploration, line 110 – 111: “Furthermore, higher temperature not only reduced reaction time but also enhanced the efficiency (entry 14)”. The Table 1 entry shows the reaction run at room temperature.”
Response: This is a typo. The word “rt” has been replaced by “40”.
- “Materials and Methods. Since none of the NMRs were taken in deuterated DMSO, remove that sentence from the general description.”
Response: As compound 3r was analyzed in d6-DMSO, we suggest keeping this sentence.
- “General comments: the entire manuscript needs a careful re-reading and editing as there are several sentences that are unclear and a number of misspellings. Some of these have been noted above, but not all have been flagged.”
Response: As suggested, the entire manuscript has been proofread carefully and then English has been polished. Here are changes are made in the revised manuscript.
In line 13, “2-napthols” has been corrected to “2-naphthols”. In line 42, “The reactivity transforms into an unprecedented synthetically powerful Mannich-Diels-Alder cascade manifold for the construction of 2H-benzo[h]chromenes.” has been replaced by “The reactivity is harnessed for developing a synthetically powerful Mannich-Diels-Alder cascade manifold for the construction of 2H-benzo[h]chromenes.”
In line 80, “Because of the better leaving ability of arylamines compared to aliphatic amines,” has been replaced by “Because of the better leaving ability of arylamines compared with aliphatic amines,”.
In line 132, “The reaction was perform at room temperature.” has been replaced by “The
reaction was performed at room temperature.”
In line 160, “rescionol” has been replaced by “resorcinol”.
In line 116, “when the reaction performed was in dry DCM under by Ar ” has been replaced by “when the reaction was performed in dry DCM under Ar atmosphere”.
Reviewer 2 Report
In this manuscript, the authors reported a simple arylamine-catalyzed synthesis of substituted 2H-benzo[h]chromenes from the reaction of 1- and 2-napthols and trans-cinnamaldehydes. The authors proposed that an important intermediate of o-QMs was generated and underwent the electrocyclization to provide the final product. This is a rather novel and interesting findings which is the most striking feature of the work. However, from this reviewer’s point of view, some additional work should be carried out to corroborate the statement. The authors propose the reaction first undergoes the Mannich-type reaction to give intermediate 5, which is converted to the crucial o-QMs intermediate. However, one might also conceive the reaction first undergoes the O-Michael addition and then the aniline-promoted F-C reaction/elimination reaction to afford the final product. The authors should rule out the possibility by carrying out some control experiments. For example, the reaction of 1- or 2-napthol, aldehyde and aniline to see if the corresponding analog of intermediate 5 (without double bond) could be generated under the conditions. It is also suggested the HRMS analysis of the reaction mixture be carried out to corroborate the generation of o-QMs.Author Response
See response in Response Letter

Reviewer 3 Report
In attention to manuscript molecules-1202113 entitled “Facile Synthesis of 2H-Benzo[h]chromenes via an Arylamine-catalyzed Mannich-cyclization Cascade Reaction”, the authors report a Mannich-cyclization reaction using arylamines as a catalyst in a simple and effective procedure for the synthesis of substituted 2H-benzo[h]chromenes. Overall, this is a clear, concise, and well-written manuscript. I consider that the present work can be accepted for publication in Molecules.
Some observations:
- Page 1 – line 10: Abstract: “… arylamine-catalyzed Mannich-Cyclization cascade reaction is developed…”
- Page 2 – line 44: Scheme 1: "X" (substituent) that appears in compound 2 does not appear in any other structure.
- Page 2 – line 82: Replace “trans-cinnimaldehydes” by “trans-cinnamaldehydes.”
- Page 3 – Table 1: “… 1-naphthol (2a) with trans-cinnamaldehyde (1a).”
- Page 3: I suggest mentioning in the text the formation of the chiral center in the compounds 3.
- Page 10 - line 396: Replace “trans-cinnimaldehydes” with “trans-cinnamaldehydes.”
Author Response
Reviewer 3
- “Page 1 – line 10: Abstract: “… arylamine-catalyzed Mannich-Cyclization cascade reaction is developed…”
Response: As suggested, word “reaction” has been added in this sentence.
- “Page 2 – line 44: Scheme 1: "X" (substituent) that appears in compound 2 does not appear in any other structure.”
Response: As suggested, Scheme 1 has been corrected
- “Page 2 – line 82: Replace “trans-cinnimaldehydes” by “trans-cinnamaldehydes.””
Response: As suggested, “trans-cinnamaldehydes.” has been replaced by “transcinnimaldehydes”.
- “Page 3 – Table 1: “… 1-naphthol (2a) with trans-cinnamaldehyde (1a).””
Response: As suggested, “1-naphthol with trans-cinnamaldehyde” has been replaced by “1-naphthol (2a) with trans-cinnamaldehyde (1a)”.
- “Page 3: I suggest mentioning in the text the formation of the chiral center in the compounds 3.”
Response: Although the chiral center is formed, this is a non-asymmetric synthesis using an achiral catalyst. So it is not necessary to mention the formation of the chiral center.
- “Page 10 - line 396: Replace “trans-cinnimaldehydes” with “trans-cinnamaldehydes.””
Response: As suggested, “trans-cinnimaldehydes” has been replaced by “transcinnamaldehydes.
Round 2
Reviewer 2 Report
The authors have carried out some control experiments to address the concerning issue raised by this reviewer. However, the authors only use HRMS to postulate the formation of intermediates 5a and 6a. This is obviously not sufficient enough. It is suggested that the authors have these intermediates, at least intermediate 6a being isolated, purified and fully characterized. Furthermore, the reaction in equation C is rather confusing as the reaction conditions are identical with that in equation B. Furthermore, intermediate 6c should not exist in its ketone form, while the stable enol form can not react with the aniline any more.
Author Response
See response in attachment
